# Inhibition of LPS-Induced Skin Inflammatory Response and Barrier Damage via MAPK/NF-κB Signaling Pathway by *Houttuynia cordata* Thunb Fermentation Broth

**DOI:** 10.3390/foods13101470

**Published:** 2024-05-10

**Authors:** Zixin Song, Jiaxuan Fang, Dongdong Wang, Yuncai Tian, Yuhua Xu, Ziwen Wang, Jiman Geng, Changtao Wang, Meng Li

**Affiliations:** 1School of Light Industry Science and Engineering, Beijing Technology & Business University, Beijing 100048, China; 18801173697@163.com (Z.S.); 18515626356@163.com (J.F.); zwwang727@163.com (Z.W.); gengjiman@163.com (J.G.); wangct@th.btbu.edu.cn (C.W.); limeng@btbu.edu.cn (M.L.); 2Shanghai AZ Science & Technology Co., Ltd., Shanghai 201100, China; tianyuncai@shzchzp.wecom.work (Y.T.); huahua2125@126.com (Y.X.)

**Keywords:** *Houttuynia cordata* Thunb, LPS, skin inflammation, skin barrier damage, MAPK/NF-κB pathway

## Abstract

*Houttuynia cordata* Thunb is rich in active substances and has excellent antioxidant and anti-inflammatory activity. Scanning electron microscopy and gel permeation chromatography were used to analyze the molecular characteristics of the fermentation broth of *Houttuynia cordata* Thunb obtained through fermentation with *Clavispora lusitaniae* (HCT-f). The molecular weight of HCT-f was 2.64265 × 10^5^ Da, and the polydispersity coefficient was 183.10, which were higher than that of unfermented broth of *Houttuynia cordata* Thunb (HCT). By investigating the active substance content and in vitro antioxidant activity of HCT-f and HCT, the results indicated that HCT-f had a higher active substance content and exhibited a superior scavenging effect on 2,2-diphenyl-1-picrylhydrazyl radicals and hydroxyl radicals, with IC50 values of 11.85% and 9.01%, respectively. Our results showed that HCT-f could effectively alleviate the increase in the secretion of inflammatory factors and apoptotic factors caused by lipopolysaccharide (LPS) stimulation, and had a certain effect on repairing skin barrier damage. HCT-f could exert an anti-inflammatory effect by down-regulating signaling in the MAPK/NF-κB pathway. The results of erythrocyte hemolysis and chicken embryo experiments showed that HCT-f had a high safety profile. Therefore, this study provides a theoretical basis for the application of HCT-f as an effective ingredient in food and cosmetics.

## 1. Introduction

Research on natural plants and foods with physiologically and pharmacologically active effects has been a hot topic in recent years. In addition to serving as food to meet people’s daily needs, traditional natural plants can also be utilized as raw materials for medicines and cosmetics [1]. Finding plant components with exceptional antioxidant, anti-inflammatory, and other efficacious effects is becoming more and more crucial as the market for safe and environmentally friendly cosmetic ingredients grows.

*Houttuynia cordata* Thunb is a perennial herb that was first cultivated in the mountainous regions of China, Japan, Korea, and Southeast Asia [2]. As a medicinal and food plant, it has a long history of application with a high degree of safety [3]. *Houttuynia cordata* Thunb is rich in a variety of bioactive components such as polyphenols, vitamins, and trace elements. It exhibits various bioactive effects, including anti-inflammatory, antibacterial, anti-cancer and antiviral properties [4,5,6,7]. It has been demonstrated that the ethanol extract of *Houttuynia cordata* Thunb can successfully reduce the oxidative damage to the liver and atherosclerosis brought on by a high-fat diet in rats [8]. Fang demonstrated that *Houttuynia cordata* Thunb extract could decrease the initiation of neuronal inflammation and apoptosis induced by propofol by enhancing neuronal activity [9]. *Houttuynia cordata* Thunb has shown good bacteriostatic activity against bacteria such as Staphylococcus aureus and Casecoccus [10].

*Clavispora lusitaniae* is an ester-producing yeast with a wide range of applications in enhancing the flavor and quality of white wines [11]. It can tolerate simulated gastric juices at pH 2.5 for 2 h and can produce extracellular polysaccharides during its growth [12]. The food products fermented by *Clavispora lusitaniae* are safer and richer in nutrients [13]. Recent studies have demonstrated that *Clavispora lusitaniae* JARR-1 has the capability to produce erythritol and ethanol [14].

Microbial fermentation technology refers to the utilization of microorganisms, under suitable conditions, to convert raw materials through specific metabolic pathways into materials required by humans. It mainly includes two categories: aerobic fermentation and anaerobic fermentation. This technology has a wide range of applications in the chemical industry, pharmaceutical industry, and other sectors [15,16,17]. The active ingredients of plants are primarily distributed in the cell, but the cell wall can hinder the release of these active ingredients. The microbial fermentation process generates enzymes such as cellulase and pectinase, which aid in breaking down the cell wall, facilitating the better release of active ingredients [18,19,20].

The skin is the body’s natural protective barrier. When exposed to foreign substances, it can not only effectively resist the invasion of pathogens but also maintain the body’s balance and prevent the loss of useful substances [21]. When the skin is stimulated by external harmful factors, its own barrier function will be compromised. This, in turn, triggers a series of immune responses that help maintain the skin’s structure and function by modifying intercellular molecular signaling processes and facilitating matrix remodeling [22]. Low doses of nitric oxide (NO) have many functions, such as promoting wound healing and exhibiting antimicrobial effects against bacterial pathogens. The expression of the inducible isoform of NOS (iNOS) predicts inflammation. The expression of iNOS promotes the production of large amounts of NO, which in turn promotes various acute and chronic inflammatory conditions [23]. Lipopolysaccharides (LPS), complexes of lipids and polysaccharides, consist of three main components that can trigger a cascade of immune-stimulatory responses and toxic pathophysiological activities in the body. Studies have shown that LPS induce oxidative stress by activating vascular endothelial cells, leading to the synthesis and release of NO and ROS, and promoting the secretion of inflammation-associated factors. This process can result in inflammatory damage to endothelial cells [24]. Wang and his team found that aloe extracts inhibited the activation of ERK and JNK caused by LPS-induced injury, which is one of the reasons for its anti-inflammatory activity [25].

Several studies have demonstrated that numerous plants and their extracts possess inhibitory effects on the activation of the nuclear factor-κB (NF-κB) signaling pathway [26]. This pathway is one of the mechanisms through which these plants exert their anti-inflammatory effects. When IκB, which binds to NF-κB, is phosphorylated or degraded, NF-κB is activated, leading to its translocation into the nucleus and promoting the release of inflammatory factors. This, in turn, triggers a series of inflammatory responses [27]. The mitogen-activated protein kinase (MAPK) signaling pathway can also regulate inflammatory responses by up-regulating cytokine expression [28]. It has been shown that *Houttuynia cordata* Thunb extract exhibits anti-inflammatory activity. The mechanism behind this effect may involve the inhibition of the activation of the MAPK/NF-κB pathway, which is closely associated with the inflammatory response [29].

The anti-inflammatory effects of *Houttuynia cordata* Thunb extracts have been studied by many scholars. However, the role of *Houttuynia cordata* Thunb fermentation obtained via *Clavispora lusitaniae* (HCT-f) in an LPS-induced HaCaT cell inflammation model has not been reported. In this study, HCT-f was prepared, and the active substance content, molecular weight size, apparent morphological characteristics, in vitro antioxidant activity, ROS regulation ability, effect on apoptosis, and modulation of inflammatory factors were determined (Figure 1). This provided a theoretical basis and data support for the application of HCT-f.

## 2. Materials and Methods

### 2.1. Materials and Chemicals

*Houttuynia cordata* Thunb (Hubei Enshi, China); *Clavispora lusitaniae* (strain number: GDMCC64135, Guangdong Microbial Culture Collection Center); human immortalized keratinocytes (HaCaT) cells (National Experimental Cell Resources Sharing Service Platform, 4201HUM-CCTCC00106); hydrogen peroxide (H_2_O_2_), salicylic acid, ferrous sulfate (FeSO_4_), and 2,2-diphenyl-1-picrylhydrazyl (Sinopharm Chemical Reagent Co., Ltd., Shanghai, China); ABTS and FRAP total antioxidant capacity assay kits (Beyotime Biotechnology, Shanghai, China); BCA protein assay kit (Biorigin Inc., Beijing, China); ROS (reactive oxygen species) assay kit (Beyotime Biotechnology, Shanghai, China); cell-counting kit-8 (Biorigin Inc., Beijing); DMEM (Dulbecco’s modified Eagle medium) and PBS (phosphate-buffered saline) (Gibco Inc., Billings, MT, USA); 96-well plate (Corning, NY, USA); ELISA kits (Nanjing Jiancheng Bioengineering Research Institute, Nanjing, China).

### 2.2. Preparation of HCT and HCT-f

*Houttuynia cordata* Thunb was dried and crushed, passed through a 100-mesh sieve, added to distilled water (1:100), mixed well, sterilized at 121 °C for 20 min in an autoclave (BXM-30R Vertical Pressure Steam Sterilisation Kettle, Shanghai Boxun Industrial Co., Ltd., Shanghai, China), and then cooled to room temperature. *Clavispora lusitaniae* was inoculated and placed on a shaker (DQHZ-2001 A Full Temperature Shock Incubator, Taicang Huamei Biochemical Instrument Factory, Taicang, China) at 180 r/min, fermented at 28 °C for 48 h. It was then transferred into a high-pressure sterilization pot (Sigma, Steinheim, Germany) for sterilization (121 °C, 20 min) and centrifuged at 4800 r/min for 30 min, and the resulting supernatant was identified as the fermentation broth of *Houttuynia cordata* Thunb (HCT-f).

The preparation process for the unfermented broth of *Houttuynia cordata* Thunb (HCT) and HCT-f was essentially the same, except that it was not inoculated with *Clavispora lusitaniae*.

### 2.3. GPC (Gel Permeation Chromatography) and SEM (Scanning Electron Microscope)

The changes in molecular weight size of both HCT and HCT-f samples were determined using gel permeation chromatography (GPC). The samples were coated with gold on the sample surface and observed using the FEI Nova Nano SEM 450 instrument (Thermo Fisher Scientific, Waltham, MA, USA).

### 2.4. Determination of HCT-f Content

#### 2.4.1. Total Sugars

The total sugar content of HCT-f was determined by using a total sugar content assay kit (Suzhou Grace Biotechnology Co., Suzhou, China). A master standard solution of glucose at a concentration of 1 mg/mL was prepared and then diluted to create six different concentration gradients: 0, 0.2, 0.4, 0.6, 0.8, and 1 mg/mL. These solutions were utilized to construct the standard curve. The specific experiments were carried out following the instructions provided in the kit.

#### 2.4.2. Polysaccharides

The content of reducing sugar in HCT-f was determined by using a reducing sugar assay kit (Suzhou Grace Biotechnology Co., Suzhou, China). A master standard solution of glucose at a concentration of 1 mg/mL was prepared and then diluted to create six different concentration gradients: 0, 0.2, 0.4, 0.6, 0.8, and 1 mg/mL. These solutions were utilized to construct the standard curve as follows: Add 100 μL of different concentrations of samples and standards sequentially into the test tube (add an equal amount of distilled water to the blank tube). Then, add 100 μL of reagent I, mix uniformly, and heat it in a water bath (Beijing Jingke Huarui Instrument Co., Beijing, China) at 95 °C for 10 min. Remove it and immediately cool it down to room temperature with cold water. Add 1 mL of distilled water, mix it uniformly, and measure the absorbance value at 500 nm. Finally, use the standard curve to determine the amount of reducing sugar in the samples.
Polysaccharide content = total sugar content − reducing sugar content(1)

#### 2.4.3. Total Proteins

The total protein content in HCT-f was determined using a BCA protein content assay kit (Biorigin Inc., BN27109, Beijing, China), and the specific experimental procedures were performed according to the instructions.

#### 2.4.4. Total Phenols

The total phenol content of the samples was determined using the forintol method [30]. Different concentrations of gallic acid standard solution and Na_2_CO_3_ (sodium carbonate, Sinopharm Chemical Reagent Co., Ltd., Shanghai, China) solution with a volume fraction of 26.7% were prepared. A total of 200 μL of gallic acid standard solution of different concentrations and the sample to be tested were added to a 2 mL EP tube. Next, 200 μL of deionized water, 100 μL of doubly diluted forintol reagent and 300 μL of 26.7% Na_2_CO_3_ solution were added sequentially, deionized water was added to reach 2 mL, and the reaction was carried out for 2 h. OD760 was measured by using an enzyme calibrator (Tecan Trading Co., Ltd., Männedorf, Switzerland), and the absorbance of the samples was brought into the standard curve for calculation.

### 2.5. Determination of HCT-f Antioxidant Capacity

#### 2.5.1. DPPH Scavenging Assay

The determination of the scavenging capacity of the samples for DPPH (2,2-diphenyl-1-picrylhydrazyl) radicals was carried out with reference to the method of Su and her team [31]. The DPPH solution was prepared by dissolving 8 mg of DPPH in 100 mL of anhydrous ethanol. Six concentration gradients were set for each sample to be tested. The sample tube (A_1_) was filled with 1 mL of various concentrations of the sample and 1 mL of DPPH. The blank tube (A_3_) was filled with 1 mL of different concentrations of the sample and 1 mL of deionized water, while the control tube (A_2_) contained 1 mL of deionized water and 1 mL of the DPPH solution (Table 1). After 30 min of reaction protected from light, OD517 was determined.
DPPH free radical scavenging rate % = [(A_2_ − A_1_ + A_3_)/A_2_] × 100%(2)

#### 2.5.2. Hydroxyl Radical Scavenging Assay

The scavenging capacity of the samples for hydroxyl radicals was determined using the salicylic acid method [31]. An 8 mmol/L FeSO_4_ (ferrous sulfate) solution, a 3 mmol/L salicylic acid solution, and a 0.02 mol/L H_2_O_2_ (hydrogen peroxide) solution were prepared. To the blank tube (A_0_), 0.3 mL of FeSO_4_ solution, 1 mL of salicylic acid, 1.45 mL of deionized water, and 0.25 mL of H_2_O_2_ were added. The sample tube (A_1_) was filled with 0.3 mL of FeSO_4_ solution, 1 mL of salicylic acid, 1 mL of various concentrations of the sample, 0.45 mL of deionized water, and 0.25 mL of H_2_O_2_. The sample background tube (A_2_) was filled with 0.3 mL of FeSO_4_ solution, 1 mL of various concentrations of the sample, 1.45 mL of deionized water, and 0.25 mL of H_2_O_2_ (Table 2). The reaction was carried out at 37 °C for 1 h. The speed of the centrifuge was set to 4000 r/min and centrifuged for 10 min, and the OD510 was determined.
Hydroxyl free radical scavenging rate % = [(A_0_ − A_1_ + A_2_)/A_0_] × 100%(3)

#### 2.5.3. Determination of Total Antioxidant Capacity

The total antioxidant capacity of HCT-f was determined using the ABTS test kit (Beyotime Biotechnology, S0119, Shanghai, China) and FRAP test kit (Beyotime Biotechnology, S0116, Shanghai, China). Refer to the instructions for specific experimental steps.

### 2.6. Cell Culture and Viability Experiment

HaCaT cells were rapidly lysed in a water bath at 37 °C, and cultured in a T25 culture flask with 5 mL of medium (10% fetal bovine serum and 1% penicillin–streptomycin). The culture flask was incubated in a incubator (37 °C and 5% CO_2_), and when the amount of cell apposition reached 80–90%, the cells were digested and centrifuged at 1500 r/min for 5 min for the next round of passaging culture.

HaCaT cells at a good growth state were selected and 100 μL of cell suspension was inoculated into each well of a 96-well plate (1 × 10^4^ cells). The 96-well plates inoculated with cells were incubated for 12 h (no cells were added to the cell-free group), and 100 μL of serum-free medium was added to the control group (group C) and cell-free group. Then, 100 μL of a 200 μg/mL lipopolysaccharide (LPS) solution was added to each of the remaining wells. The plate was incubated for 24 h; next, 100 μL of serum-free medium was added to the control group (group C) and cell-free group, 100 μL of 200 μg/mL lipopolysaccharide (LPS) solution was added to each of the remaining wells, and the plate was incubated for 24 h. Additionally, 100 μL of samples at varying concentrations were added to the sample group. The plates were then incubated for 24 h in the incubator. The effect of samples on cell viability was assessed using the CCK8 method (Biorigin Inc., BN15201, Beijing, China). The plates were washed twice with Phosphate buffered saline (PBS). Then, 100 μL of serum-free medium and 10 μL of CCK8 reagent were added to each well. The plates were incubated in the incubator at 37 °C for 2–4 h. Absorbance values at 450 nm were measured, and cell viability was calculated using the formula below.
Cell viability = (absorbance value of the sample group − absorbance value of the cell free group)/(absorbance value of the group C − absorbance value of the cell free group) × 100%(4)

### 2.7. ROS Detection

In a 6-well plate, each well was inoculated with 2 mL of cell suspension (5 × 10^5^ cells) and placed in incubator for 24 h. Next, 2 mL of serum-free medium was added to group C, and 2 mL out of 200 μg/mL of LPS was added to both group M and the sample group. The plate was incubated in an incubator for 24 h, and 2 mL of different samples was added to the sample group, and 2 mL of serum-free medium was added to both groups C and M. The plate was incubated in an incubator for 24 h. The effect of different samples on ROS levels was assayed by using a reactive oxygen detection kit (Beyotime Biotechnology, S0033S, Shanghai, China). The fluorescence intensity of ROS in the cells was observed using a fluorescence-inverted microscope, photographed, and recorded.

### 2.8. The Effect of HCT-f on Apoptotic Cell Death

#### 2.8.1. Ca^2+^ Concentration Measurement

In a 6-well plate, each well was inoculated with 2 mL of cell suspension (5 × 10^5^ cells) and placed in an incubator for 24 h. Changes in intracellular Ca^2+^ concentration levels were observed by the specific binding of a Fluo-4, AM ester Ca^2+^ fluorescent probe to Ca^2+^. Firstly, a 4 μM Fluo-4, AM ester working solution was prepared, the 6-well plate was washed with PBS for 3 times, and 1 mL of Fluo-4, AM ester working solution was added to each well. It was then incubated at 37 °C for 20 min in an incubator maintained at constant temperature and humidity. The Fluo-4, AM ester working solution was discarded, the plate was washed 3 times, 1 mL PBS buffer was added, and incubated at 37 °C for 10 min. The changes in intracellular Ca^2+^ concentration were observed using a fluorescence-inverted microscope, and the experimental results were recorded.

#### 2.8.2. Mitochondrial Membrane Potential Measurement

In a 6-well plate, each well was inoculated with 2 mL of cell suspension (5 × 10^5^ cells) and placed in an incubator for 24 h. The mitochondrial membrane potential of HaCaT cells was measured using the Beyotime enhanced mitochondrial membrane potential detection kit as follows: Take 50 μL of JC-1 (200X) and add 8 mL of ultrapure water. Mix well to fully dissolve, then add 2 mL of JC-1 staining buffer (5X) and mix well to prepare the staining working solution. The specific operation was conducted following the instruction manual, and the experimental results were observed and recorded using a fluorescence microscope (CKX53, Olympus LS, Tokyo, Japan).

### 2.9. Enzyme-Linked Immunosorbent Assay (ELISA)

A 6-well plate was lined with 50 × 10^4^ cells per well and incubated. The cell supernatant was collected in 2 mL centrifuge tubes, and the cells at the bottom of the 6-well plate were lysed to collect the cell lysate. Both cell supernatant and lysate were centrifuged at 10,000 r for 5 min, and the precipitate was discarded and stored at −20 °C for subsequent experiments. The experiments were performed according to the corresponding instructions of the IL-6, IL-8, iNOS, TNF-α, AQP3, FLG, LOR, Caspase-14, Caspase-3, and Caspase-9 assay kits.

### 2.10. qRT-PCR

Primer sequences were designed using PrimerExpress software (version 3.0). Total RNA extraction and cDNA synthesis were performed using Trizol reagent and a reverse transcription kit (Biorigin, BN12028, Beijing, China). Refer to the Fast Super EvaGreen^®^ qPCR Master Mix kit (Biorigin, BN12008, Beijing, China) instructions for subsequent experiments. The reverse transcription system is shown in Table 3. Primer sequences are shown in Table 4. The PCR reaction system is shown in Table 5.

### 2.11. Safety Determination

#### 2.11.1. Rabbit Erythrocyte Hemolysis Assay (RBC)

Fresh rabbit blood was centrifuged at 1500× *g* for 10 min, and the precipitate and PBS were taken and diluted in a 1:49 ratio. Blood washing was completed when 0.5 mL of cell suspension was added with 4.5 mL of deionized water and the OD541 was 0.5 ± 5%. The washed blood and the samples to be tested were mixed well in the ratio of 1:3, incubated for 60 min on a shaking bed, and the incubation was terminated by centrifugation for 10 min at 10,000× *g*. The absorbance value at 560 nm was measured. The above experimental procedure was repeated with PBS and deionized water instead of the sample as the negative control (NC) and with the positive control (PC) of the sample.
Hemolysis = (sample OD_560_ − NC OD_560_)/(PC OD_560_ − NC OD_560_) × 100%(5)

#### 2.11.2. Chicken Embryo Chorionic Allantoic Membrane Eye Irritation Test (HET-CAM)

Fertilized eggs were incubated in an incubator for 10 days (38 °C, 60–70% humidity), and the position of the air chambers was observed under a light source. The eggshell in the upper part of the air chamber was removed with tweezers. The eggshell membrane was moistened with 0.9% NaCl solution and torn off with forceps to expose the blood vessels. Next, 300 μL of 0.9% NaCl solution, 0.1 mol/L NaOH solution, and the samples were added directly to the exposed blood vessels, and the blood vessel damage was observed for 3 min, which served as the negative control group, the positive control group, and the sample group, respectively, and the time of bleeding, coagulation, and dissolution of the blood vessels were recorded. The samples were evaluated for eye irritation using the endpoint evaluation method (Table 6 and Table 7).

### 2.12. Data Analysis

All experiments were repeated three times. Data were analyzed and plotted using GraphPad Prism 8 (GraphPad Software, Inc., San Diego, CA, USA), the significance of the data was analyzed using conventional one-way analysis of variance (ANOVA), and the data were processed and IC50 calculated using SPSS software 23. Differences were considered significant when *p* < 0.05.

## 3. Results

### 3.1. Characterization of the Microscopic Morphology of HCT-f

The molecular weights (MWs) of HCT and HCT-f are shown in Table 8. The average molecular weight of HCT was 1.02758 × 10^5^ Da, and that of HCT-f was 2.64265 × 10^5^ Da. The molecular weight was increased after fermentation, which might be due to the fact that microorganisms produce enzymes to promote the rupture of the plant cell wall in the process of fermentation, which promotes the release of the macromolecular active substances. The polydispersity coefficient (Mw/Mn) of HCT-f is 183.10, which is higher than that of HCT, indicating a more complex molecular composition. The morphological characteristics of *Houttuynia cordata* Thunb under different levels of precision are shown in Figure 2. It can be seen that HCT-f has better structural homogeneity than HCT, and under the same precision, the surface morphological structure of HCT-f presents more pores and is rougher, indicating that HCT-f has better absorbability and adhesion.

### 3.2. Active Substance Content and In Vitro Antioxidant Activity of Houttuynia cordata Thunb

The various bioactive effects of *Houttuynia cordata* Thunb, such as anti-inflammatory, antioxidant, and antimicrobial, are closely related to its rich content of bioactive substances. We determined the changes in the bioactive substance content and in vitro antioxidant activity of *Houttuynia cordata* Thunb before and after fermentation (Figure 3).

As can be seen in Figure 3A, HCT and HCT-f were rich in total sugars, polysaccharides, and total proteins, and showed reduced polyphenolic compounds. After fermentation, a significant increase in the content of all four bioactive substances occurred. The total sugar content and polysaccharide content of HCT-f were both significantly higher compared to HCT, at 4.52 and 2.92 mg/mL, respectively.

Figure 3B showed that the in vitro antioxidant activity of HCT-f was superior to that of HCT. The scavenging ability of HCT-f with a volume fraction of 25% on hydroxyl radicals could reach more than 90%, which exceeded that of HCT with a volume fraction of 100%, and there was little difference in the scavenging effect of different concentrations of HCT and HCT-f on DPPH radicals. The smaller IC50 represents the better scavenging ability of the sample on free radicals, and it can be seen in the figure that the HCT-f with a volume fraction of 11.85% could scavenge 50% of DPPH radicals. The experimental results of the total antioxidant capacity assay showed that the scavenging ability of HCT-f and HCT on ABTS radicals did not differ much, and the reducing ability of HCT-f on Fe^2+^ was 1.67 times that of HCT, which could reach 1.11 mM Trolox equivalent. 

In conclusion, we can see that after fermentation, the content of bioactive substances and the in vitro antioxidant activity of *Houttuynia cordata* Thunb were increased, which suggests that compared with HCT, HCT-f may have a higher application value.

### 3.3. Effect of HCT-f on Cell Viability

The CCK8 assay was utilized to assess whether HCT-f (0.625–20%) inhibited cell viability in a volume fraction-dependent manner. Figure 4A demonstrates that, compared to the blank control group C, varying volume fractions of HCT-f had a specific impact on HaCaT cells in promoting cell proliferation, with HCT-f showing a superior promotion of cell proliferation. To further investigate the impact of HCT-f treatment on the proliferation of HaCaT cells following LPS-induced injury, the LPS-injured HaCaT cells were exposed to varying concentrations of HCT-f. The results showed (Figure 4B) a significant decrease in cell viability in cells treated with LPS alone (group M) compared to group C without LPS-induced injury (*p* < 0.001). Compared with group M, HCT-f with a volume fraction of 1.25% to 10% restored cell viability in a dose-dependent manner, suggesting that treating cells with HCT-f attenuates LPS-induced damage in HaCaT cells. For the comprehensive assessment of cell viability, status, and the potential toxic effects of a high volume fraction (20%) of HCT-f on cells, we opted to utilize HCT-f with a volume fraction of 10% for the upcoming cell experiments.

### 3.4. Effect of HCT-f on Reactive Oxygen Species (ROS) Content in Cells

Under normal circumstances, ROS, as natural byproducts of oxygen metabolism, are present at low levels in the organism. They participate in intracellular signaling and regulation as redox messengers, playing a crucial role in the cell cycle and gene expression. When the body is stimulated by exogenous stimuli such as LPS, the content of ROS will increase rapidly, surpassing the body’s natural capacity to eliminate them. This disruption will then disturb the dynamic balance between ROS production and removal in the body, causing oxidative stress. Oxidative stress can result in DNA damage, changes in protein structure and function, and harm to cell membranes.

From Figure 5B, it can be seen that the intensity of green fluorescence was significantly enhanced in group M after LPS induction compared to group C without LPS induction. The intensity of green fluorescence was weakened after treatment with both HCT and HCT-f at a volume fraction of 10%. Additionally, the effect of HCT-f was superior to that of HCT. Figure 5A shows that after LPS injury, the relative expression of ROS in group M was significantly increased. Treatment with both HCT and HCT-f reduced the generation of ROS. The inhibitory effect of HCT-f on the generation of ROS was superior to that of HCT.

### 3.5. Effects of HCT-f on Apoptotic Cell Death

Mitochondrial membrane potential can be used to assess the proper functioning of mitochondria. It is widely recognized that ROS are the primary cause of mitochondrial dysfunction [32]. An increase in ROS content can disrupt the mitochondrial membrane, altering its permeability and ultimately resulting in a decrease in membrane potential. The concentration of Ca^2+^ in vivo also plays a crucial role in maintaining the permeability of the mitochondrial membrane. An increase in the concentration of Ca^2+^ in the mitochondrial leads to elevated levels of ROS, which subsequently activate the Caspase-dependent apoptotic pathway [33]. Under normal conditions, the inner mitochondrial membrane potential is high and remains at a negative potential, while the outer membrane potential is low and remains at a positive potential. The decrease in mitochondrial membrane potential is associated with autophagy, apoptosis, and necrosis of cells [34].

When the level of Ca^2+^ increases, it promotes ATP production, leading to more free electrons being leaked to form superoxide. This results in the generation of a large amount of ROS, which in turn leads to the oxidation of unsaturated fatty acids, amino acids, and DNA damage, ultimately resulting in apoptosis [35]. The detection of changes in intracellular Ca^2+^ concentration can be used to initially assess whether the cell is about to undergo apoptosis. The Figure 6C experimental results show that after LPS-stimulated damage, the intensity of green fluorescence significantly increased, and the relative expression of Ca^2+^ underwent a significant increase compared to group C. Following treatment with HCT and HCT-f, the intensity of green fluorescence decreased to varying degrees, and the relative expression of Ca^2+^ was reduced compared to group M. The inhibitory effect of HCT-f on Ca^2+^ expression was superior to that of HCT.

As shown in Figure 6A, in group C, JC-1 aggregated in the mitochondrial matrix to form polymers, emitting strong red fluorescence. After LPS injury (group M), the membrane potential was reduced, and JC-1 existed in the cytosol in the form of monomers emitting strong green fluorescence, which induced an early apoptotic cell response. After 10% HCT and HCT-f treatments, the content of JC-1 polymer significantly increased, while the monomer content decreased significantly compared to group M. The effect of HCT-f was superior to that of HCT, and after 10% HCT-f treatment, it was equal to the undamaged state.

The alteration of mitochondrial permeability leads to the formation of apoptotic vesicles, which in turn promotes the activation of the Caspase family. Caspase-9 plays a crucial role in initiating apoptosis during the apoptotic process. Stimulation of apoptotic signaling activates the Caspase-9 precursor, which then triggers the Caspase-3 activation and initiates the Caspase cascade reaction, ultimately leading to apoptosis [36]. We further confirmed the anti-apoptotic effect of HCT-f by measuring the content and relative mRNA expression of Caspase-3 and Caspase-9, two proteins closely associated with apoptosis. The experimental results in Figure 6B show that the relative expression of both apoptosis-related proteins increased significantly after cells were stimulated with LPS. The up-regulation of the two apoptosis-related proteins was significantly reduced following treatment with HCT and HCT-f. Moreover, the anti-apoptotic effect of HCT-f was found to be superior to that of HCT.

In conclusion, it can be observed that HCT and HCT-f can inhibit LPS injury-induced apoptosis by mitigating the decrease in mitochondrial membrane potential, reducing Ca^2+^ inward flow, and suppressing the relative expression of Caspase-3 and Caspase-9 and their mRNA.

### 3.6. Effects of HCT-f on Inflammatory Factor Release and Skin Barrier Function

Dysregulation of apoptosis plays an important role in the pathophysiology of chronic inflammation and its associated symptoms. Further development of skin inflammation leads to impairment of the skin’s barrier function, making it unable to resist the invasion of foreign harmful substances, thus exacerbating the skin’s inflammatory response. As a result, the skin is caught in a vicious cycle of inflammation and barrier damage.

In Figure 7A, following LPS stimulation, the expression of IL-6, IL-8, TNF-α, and iNOS increased. The relative expression of their mRNAs also significantly increased. Subsequently, treatment with HCT and HCT-f resulted in a significant decrease in the expression of all four inflammatory factors. Notably, HCT-f demonstrated more effective inhibition of inflammatory factor expression compared to HCT.

Figure 7B shows that the content and relative expression of mRNA of three proteins, AQP3, FLG, and LOR, which are closely related to the skin barrier function, decreased significantly after LPS stimulation. Additionally, the content of Caspase-14 shearase and its relative expression of mRNA increased significantly. After treating the cells with both HCT and HCT-f samples, there was a significant increase in the content and relative expression of three proteins (AQP3, FLG, and LOR) and in the relative expression of their mRNAs. The effect of HCT-f was superior. After treating the cells with both HCT and HCT-f samples, there was a decrease in the content of Caspase-14 and the relative expression of its mRNA. The effect of HCT-f was superior.

### 3.7. Effects of HCT-f on the Transcriptional Activity of Genes in the MAPK/NF-κB Pathway

MAPKs are a group of signaling proteins that include p38 mitogen-activated protein kinase, c-Jun nh2-terminal kinase (JNK), and extracellular signal-regulated kinase (ERK) [37]. MAPKs and phosphorylated MAPK proteins can regulate NF-κB activation [38]. The activation of this signaling pathway is closely related to the development of inflammatory responses [39].

Figure 8 illustrates that following LPS-induced injury, the relative expression of mRNAs for all four genes (JNK, ERK, p38, and NF-κB) increased. Subsequently, after treatment with HCT and HCT-f, the relative expression of mRNA for all four genes decreased significantly. The inhibition of MAPK/NF-κB pathway activation by HCT-f was found to be superior to that of HCT. Our experimental results tentatively suggest that HCT-f may exert anti-inflammatory effects by inhibiting the LPS-induced activation of the MAPK/NF-κB pathway.

### 3.8. Safety of HCT-f

In order to test whether the samples were irritating, RBC test and HCT-CAM test were carried out, and the results of the experiments are shown in Figure 9. The results (Figure 9A) indicate that the hemolysis rate of HCT and HCT-f with volume fractions ranging from 0.625% to 10% was extremely low. Neither of them exhibited significant hemolysis, demonstrating the high safety of the samples. The grading of vascular irritation by HCT and HCT-f is specifically shown in Table 9. ES = 12 for the positive control of 0.1 mol/L NaOH, while ES = 0 for HCT and HCT-f. Based on the experimental results in Figure 9B, it can be observed that neither HCT nor HCT-f caused damage such as hemorrhage or coagulation of blood vessels when applied to the CAM. This indicates that the samples did not cause eye irritation. All of the experimental results mentioned above indicate that HCT-f is safe and has promising application prospects.

## 4. Discussion

It has been shown that *Houttuynia cordata* Thunb possesses a variety of biological effects such as antioxidant, antimicrobial, anti-tumor, and anti-inflammatory effects. Experiments in mice have shown that *Houttuynia cordata* Thunb polysaccharides can alleviate LPS-induced pneumonic injury by inhibiting macrophage hyperactivation [40]. It has been shown that the *Clavispora lusitaniae* produces extracellular polysaccharides during metabolism. These polysaccharides have a variety of physiological and pharmacological activities, such as anti-tumor, anti-aging and anti-infection [41]. The advantage of using microbial fermentation to prepare fermentation broth for direct applications in food and cosmetics is that no organic reagents are required in the fermentation process that may produce substances in their metabolism that are beneficial to the fermentation substrate in terms of its efficacy. The investigation of the efficacy and safety of *Houttuynia cordata* Thunb fermentation broth is a worthwhile research direction. Based on these findings, we hypothesized that *Clavispora lusitaniae Houttuynia cordata* Thunb fermentation broth (HCT-f) could play a significant role in repairing LPS-induced skin inflammation through antioxidant and anti-inflammatory pathways.

The physical properties of HCT-f were investigated using GPC and SEM. The experimental results showed that the molecular weight of HCT-f was higher than that of HCT. The difference could be attributed to the enzymes produced during the microbial fermentation process, which destroyed the plant cell wall and promoted the release of the active substances. Additionally, the surface morphology and structure of HCT-f exhibited more pores, roughness, and better structural homogeneity. These characteristics suggest that HCT-f might offer improved absorption and adhesion effects.

The results of biochemical experiments showed that a significant increase in total sugars, polysaccharides, total proteins, and total phenols in HCT-f compared to HCT. This increase is likely attributed to the production of enzymes that aid in breaking down the plant cell wall during the fermentation process. The results of the in vitro antioxidant assay showed that HCT-f exhibited better scavenging ability than HCT for DPPH radicals, hydroxyl radicals, and ABTS radicals. Additionally, its reduction of Fe^2+^ was also more effective. There is a positive correlation between the content of active substances and the physiological and pharmacological activities of plants in general. The results of the above experiments suggest that HCT-f may have a better application prospect compared to HCT.

Excessive generation of ROS, abnormal increase in Ca^2+^ concentration, disruption of mitochondrial membrane permeability, and aberrant activation of the Caspase family all contribute to the induction of apoptosis in cells [42]. Oxidative stress is a condition characterized by an imbalance between oxidative and antioxidant processes in the body, primarily triggered by the overproduction of ROS. Under normal conditions, the antioxidant enzymes in the body eliminate excessive ROS to maintain a normal ROS concentration. When cells are stimulated by LPS, they generate a significant amount of ROS, leading to the accumulation of free radicals. Excessive free radicals can directly induce DNA and organelle damage, trigger apoptosis, and stimulate the production and release of inflammatory factors [43]. It has been shown that abnormal elevation of Ca^2+^ concentration promotes apoptosis [44]. Mitochondria are essential for cellular energy metabolism and production of ROS, and are involved in various processes such as steroid metabolism, Ca^2+^ homeostasis, and free radical scavenging [45]. Mitochondrial membrane permeability transition is closely related to apoptosis. The opening of the mitochondrial permeability transition pores is induced by elevated Ca^2+^ concentration and overproduction of reactive oxygen species [46]. Apoptosis is closely related to the activation of the Caspase family. After apoptosis is induced, the initiating Caspases are recruited to specific complexes for their activation. Subsequently, the apoptotic signal is amplified in a cascade through the hydrolysis of Caspases downstream of heterologous activation [47].

We measured the changes in mitochondrial membrane potential, Ca^2+^ expression, the content of Caspase-3 and Caspase-9, and the relative expression of mRNA in HaCaT cells after LPS stimulation. The experimental results showed that after LPS stimulation, there was a significant increase in the content of ROS occurring in the cells. The sample treatment was effective at reducing the content of ROS. Following LPS-induced injury, there was a significant increase in the Ca^2+^ content and a significant decrease in the mitochondrial membrane potential in the HaCaT cells. These changes indicated that the cells were gradually progressing towards apoptosis after LPS stimulation. Subsequent to HCT-f treatment, the Ca^2+^ content was significantly reduced, the mitochondrial membrane potential was significantly reduced, and the mitochondrial membrane potential was significantly restored. This suggests that the samples may inhibit apoptosis by suppressing ROS and excessive Ca^2+^ production, as well as by repairing the reduction of mitochondrial membrane potential. The decrease in the secretion of two pro-apoptotic factors, Caspase-3 and Caspase-9, and the reduction in their relative mRNA expression following HCT-f treatment further support our experimental results.

LPS is a common inducer used to construct models of inflammation. It promotes the release of a variety of inflammatory factors and triggers a severe inflammatory response. When the skin is stimulated by external irritants, it promotes the expression of inflammatory factors, which further damages the skin’s barrier function. In this study, we found that HCT-f could attenuate the response of LPS-induced inflammation by inhibiting the expression of four pro-inflammatory cytokines (IL-6, IL-8, TNF-α, and iNOS). AQP3, FLG, LOR, and Caspase-14 are closely related to the barrier function of the skin. AQP3, which is mainly expressed in the plasma membrane of the keratinocytes of the epidermis, is known for its role in the transport of water and glycerol [48]. The cutaneous membrane sleeve forms the basis of the skin. It is created by the transmembrane proteins of keratinocytes (loricrin, filaggrin) cross-linking with each other under the action of transglutaminase [49]. Caspase-14 has been closely associated with the development of skin diseases such as photoaging and psoriasis [50]. Our study showed that HCT-f significantly promoted the expression of three proteins (AQP3, FLG, and LOR) and their mRNA, while also significantly down-regulating the expression of Caspase-14 and its mRNA. Collectively, these findings suggest that treating cells with HCT-f can significantly alleviate the inflammatory response and impaired barrier function of the skin caused by LPS injury. 

To investigate the mechanism by which HCT-f exerts anti-inflammatory and barrier repair functions, we further explored the MAPK/NF-κB pathway, as this pathway is closely related to the occurrence of inflammatory responses. When stimulated by LPS, the activated IκB kinase complex induces IκB-α phosphorylation and degradation, leading to the activation of nuclear translocation of NF-κB [51]. Our experimental results showed that stimulation with LPS activated the MAPK and NF-κB pathways. Treatment of cells with HCT-f effectively inhibited the expression of JNK, ERK, p38, and NF-κB mRNA. From this, we hypothesized that HCT-f may play an anti-inflammatory role by inhibiting the activation of the MAPK/NF-κB pathway. However, the specific effect of HCT-f on the MAPK/NF-κB pathway needs to be further verified by Western blotting experiments. The results of the RBC test and HCT-CAM test indicated that HCT-f did not cause eye irritation and was deemed safe.

Although our experimental results demonstrated that fermented *Houttuynia cordata* Thunb has a higher content of active substances, better antioxidant efficacy, and can alleviate the inflammatory damage caused by LPS stimulation by inhibiting the activation of the MAPK/NF-κB pathway, we did not exclude the influence of the *Clavispora lusitaniae* itself and the metabolites produced during the fermentation process on the experimental results. Additionally, we did not clarify the specific substances in HCT-f that exert antioxidant and anti-inflammatory effects. Although experimental results have shown that enzymes released during microbial fermentation can promote the rupture of plant cell walls, facilitating the release of active ingredients from the plant, the specific enzymes responsible for this effect during the fermentation process in our study remain unknown. Recent studies have demonstrated that the alkaloids found in *Houttuynia cordata* Thunb can exhibit anti-inflammatory effects by inhibiting nitric oxide (NO) production in RAW 264.7 cells under LPS-induced injury [52]. *Houttuynia cordata* Thunb oil could exert anti-inflammatory effects by reducing the synthesis of iNOS and TNF-α proteins in macrophages induced by LPS stimulation [53]. Our study only explored the effect of HCT-f on LPS-induced HaCaT cell damage and did not investigate its impact on other cells. We plan to conduct further research to explore the effects of HCT-f on other cell types in the future. The results of the chicken embryo test and the erythrocyte hemolysis test provide preliminary evidence that the samples are highly safe. However, this does not guarantee the same level of safety when applied to humans. In future research, we will further clarify the specific antioxidant and anti-inflammatory substances in HCT-f. We will also conduct animal and human studies to further demonstrate its efficacy and safety, aiming to realize the high-value utilization of *Houttuynia cordata* Thunb.

## 5. Conclusions

In conclusion, our experimental results indicated that compared with HCT, there was a significant increase in the content of active substances and in vitro antioxidant activity occurred in HCT-f. HCT-f could alleviate apoptosis and inflammatory responses caused by LPS. In addition, our study also indicated that HCT-f might exert the above effects by inhibiting the activation of the MAPK/NF-κB signaling pathway. The experimental results of the safety evaluation indicated that HCT-f is safe and has promising prospects for application.

## Figures and Tables

**Figure 1 foods-13-01470-f001:**
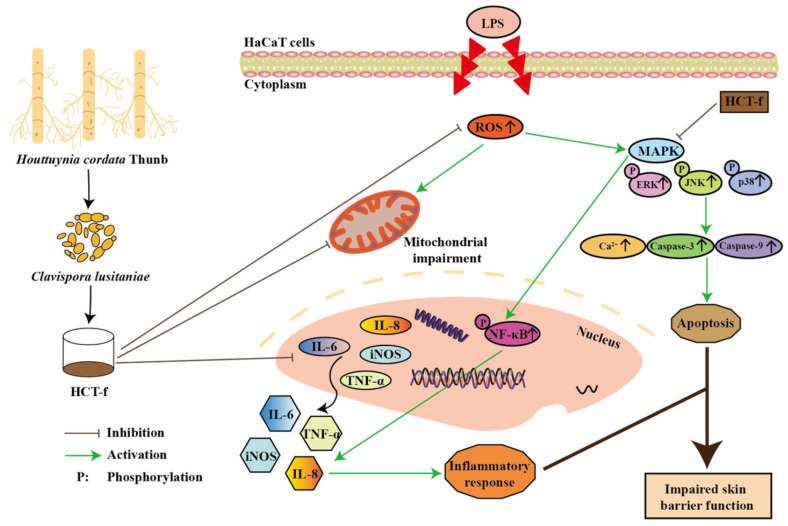
The role of HCT-f in LPS-induced injury in HaCaT cells.

**Figure 2 foods-13-01470-f002:**
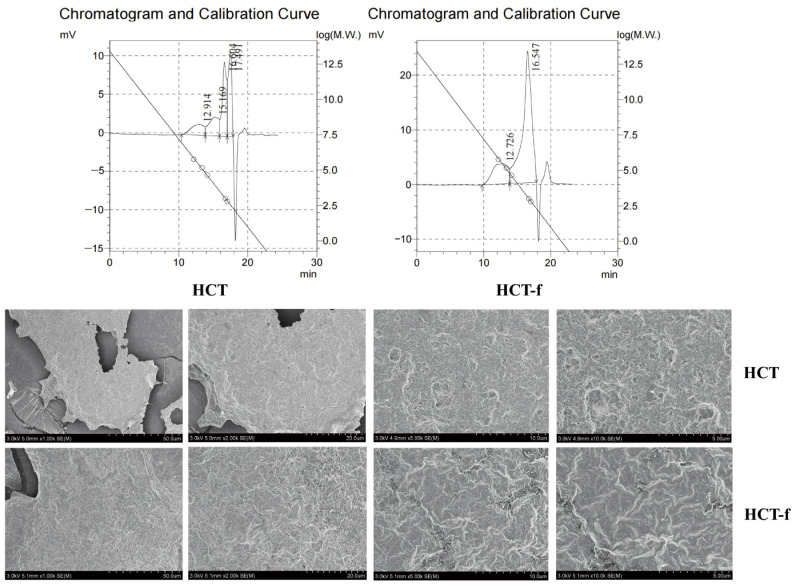
Changes in morphological and structural characteristics of *Houttuynia cordata* Thunb before and after fermentation. The magnifications of the HCT and HCT-f electron microscope images were 1000×, 2000×, 5000×, and 10,000× in that order. The diagonal lines in the figure represent standard curves. These curves were generated using a narrowly distributed set of prasinomannan-labeled samples with molecular weights of 5.8 × 10^5^, 1.46 × 10^5^, 4.42 × 10^4^, 1000 and 600 for the GPC experiments. The small arrow on the graph is the point where the peak taking starts and ends. And the circles in the graph represent the efflux times of the narrowly distributed Pullulan polysaccharide labeled groups with molecular weights of 5.8 × 10^5^, 1.46 × 10^5^, 4.42 × 10^4^, 1000 and 600.

**Figure 3 foods-13-01470-f003:**
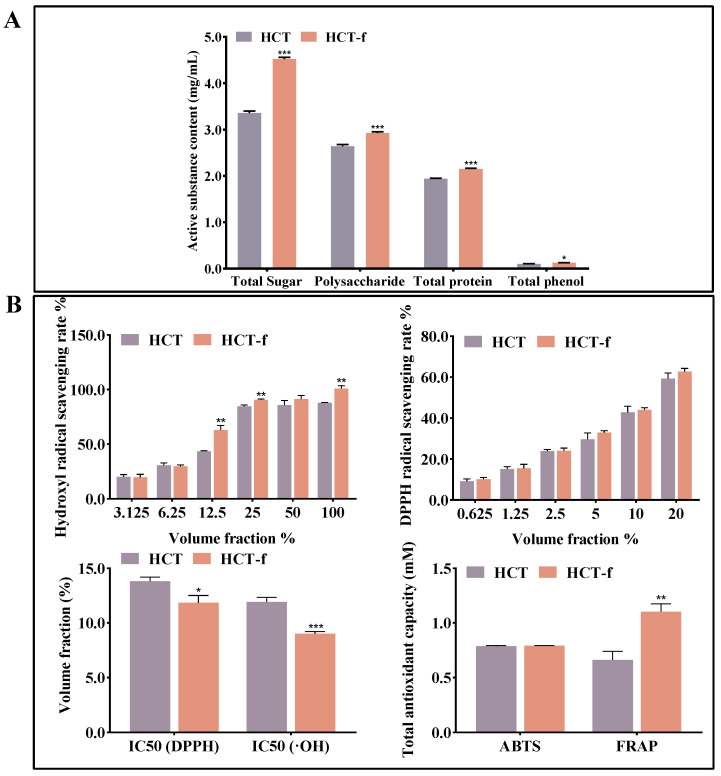
Determination of the content of bioactive substances in HCT and HCT-f and in vitro antioxidant capacity. (**A**) Determination of total sugars, polysaccharides, total proteins, and total phenols in HCT and HCT-f. (**B**) Determination of DPPH, hydroxyl radical, and ABTS radical scavenging capacity of HCT and HCT-f, and Fe^2+^ reduction capacity. One-way ANOVA was performed to determine statistical significance. * *p* < 0.05, ** *p* < 0.01, *** *p* < 0.001 vs. the HCT group.

**Figure 4 foods-13-01470-f004:**
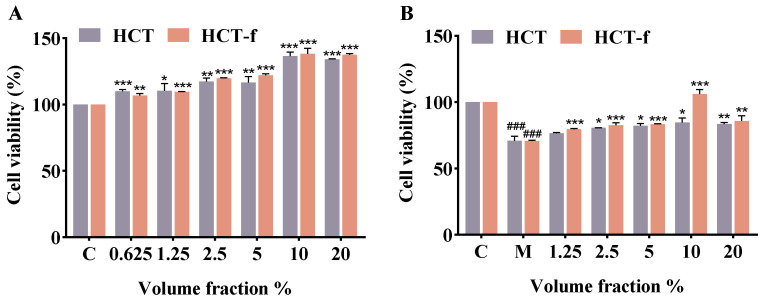
Effect of HCT and HCT-f on the viability of HaCaT cells. (**A**) Effect of 0.625–20% volume fraction of HCT and HCT-f on HaCaT cell viability. (**B**) Effect of 1.25–20% volume fraction of HCT and HCT-f on the viability of HaCaT cells with LPS-induced injury. (C: not subjected to LPS-induced injury; M: subjected to LPS-induced injury only, no sample added.) One-way ANOVA was performed to determine statistical significance. * *p* < 0.05, ** *p* < 0.01, *** *p* < 0.001 vs. the control group; ^###^
*p* < 0.001 vs. the control group, * *p* < 0.05, ** *p* < 0.01, *** *p* < 0.001 vs. the model group.

**Figure 5 foods-13-01470-f005:**
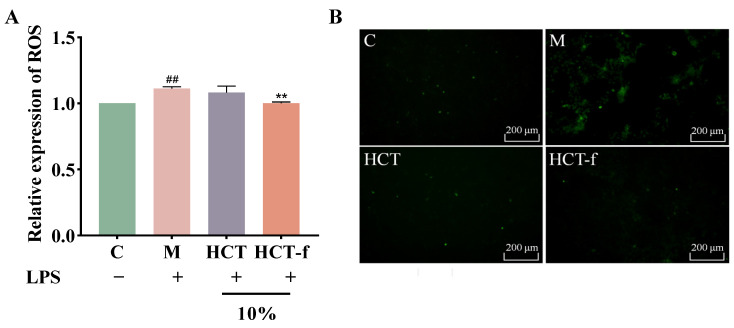
Effects of HCT and HCT-f on ROS content in LPS-induced damaged HaCaT cells. (**A**) Effects of HCT and HCT-f on the relative expression of ROS. (**B**) ROS generation (green fluorescence) was detected by fluorescence microscopy. One-way ANOVA was performed to determine statistical significance. ^##^
*p* < 0.01 vs. the control group, ** *p* < 0.01 vs. the model group.

**Figure 6 foods-13-01470-f006:**
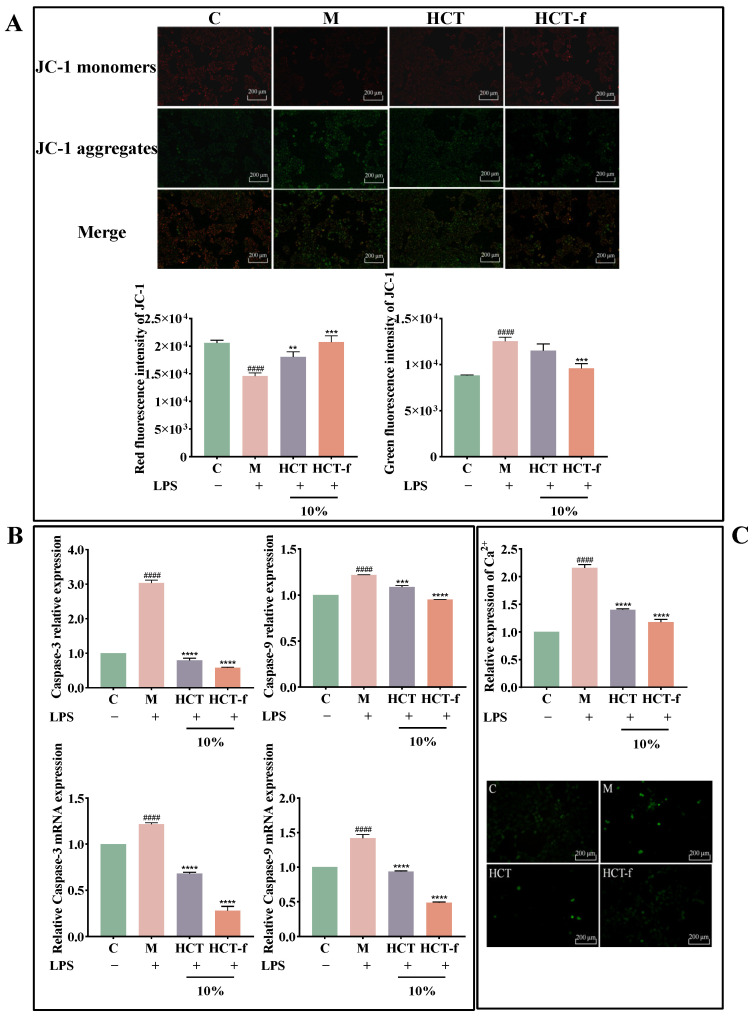
Effects of HCT and HCT-f on the apoptotic profile of HaCaT cells after LPS-induced injury. (**A**) Effects of HCT and HCT-f on the mitochondrial membrane potential in HaCaT cells. (**B**) Effects of HCT and HCT-f on the content and relative mRNA expression of two apoptotic proteins, Caspase-3 and Caspase-9, in HaCaT cells. (**C**) Effects of HCT and HCT-f on the changes in Ca^2+^ content and fluorescence intensity in HaCaT cells. One-way ANOVA was performed to determine statistical significance. ^####^
*p* < 0.0001 vs. the control group, ** *p* < 0.01, *** *p* < 0.001, **** *p* < 0.0001 vs. the model group.

**Figure 7 foods-13-01470-f007:**
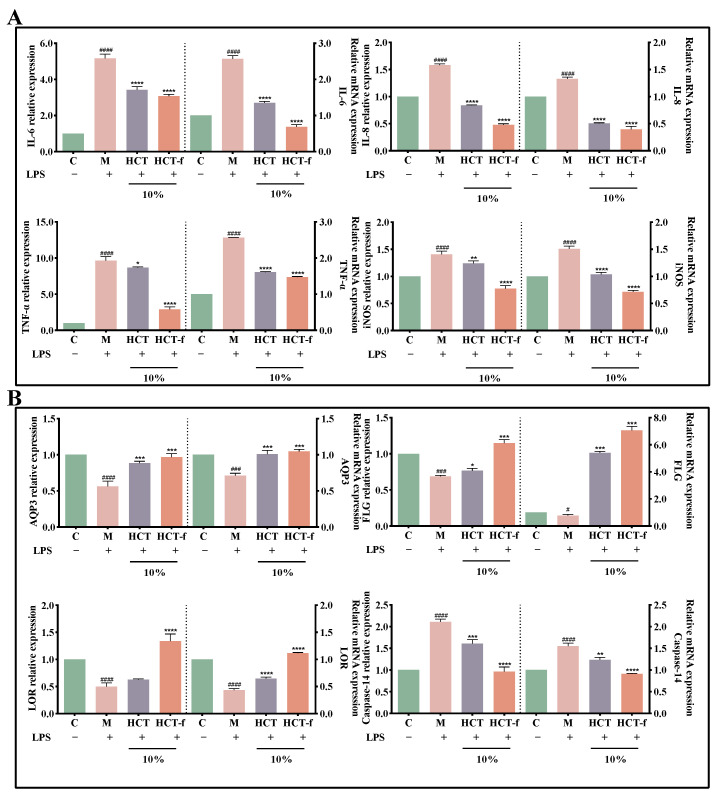
Effects of HCT and HCT-f on the release and transcript levels of inflammatory and barrier factors in LPS-induced injured HaCaT cells. (**A**) Effects of HCT and HCT-f on the release of IL-6, IL-8, TNF-α, and iNOS, as well as their mRNA expression in HaCaT cells with LPS-induced injury. (**B**) Effects of HCT and HCT-f on the release of AQP3, FLG, LOR, and Caspase-14, and their mRNA expression in HaCaT cells with LPS-induced injury. One-way ANOVA was performed to determine statistical significance. ^#^
*p* < 0.05, ^###^
*p* < 0.001, ^####^
*p* < 0.0001 vs. the control group; * *p* < 0.05, ** *p* < 0.01, *** *p* < 0.001, **** *p* < 0.0001 vs. the model group.

**Figure 8 foods-13-01470-f008:**
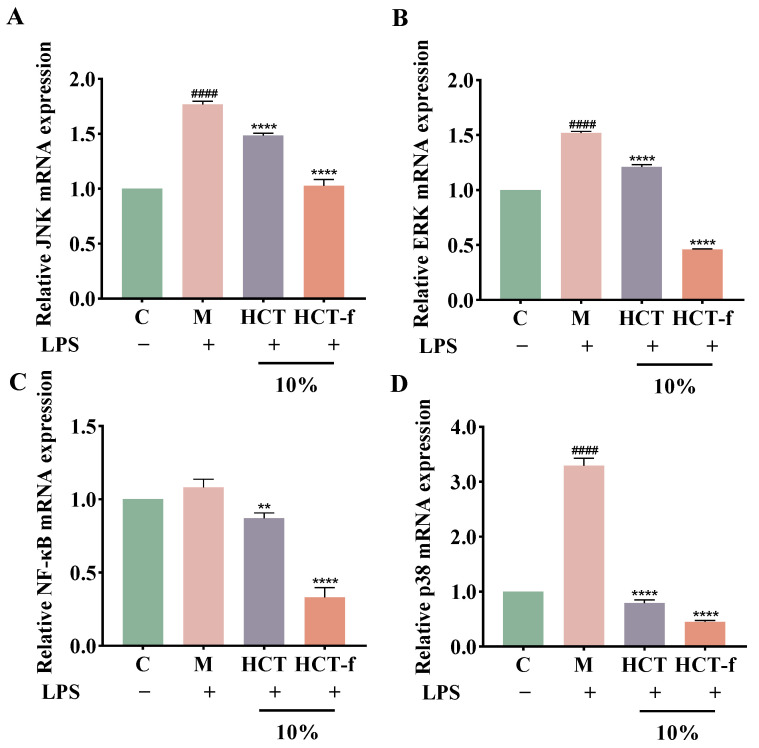
Effects of HCT and HCT-f on gene expression of the JNK, ERK, NF-κB, and p38. (**A**) JNK; (**B**) ERK; (**C**) NF-κB; (**D**) p38. One-way ANOVA was performed to determine statistical significance. ^####^
*p* < 0.0001 vs. the control group; ** *p* < 0.01, **** *p* < 0.0001 vs. the model group.

**Figure 9 foods-13-01470-f009:**
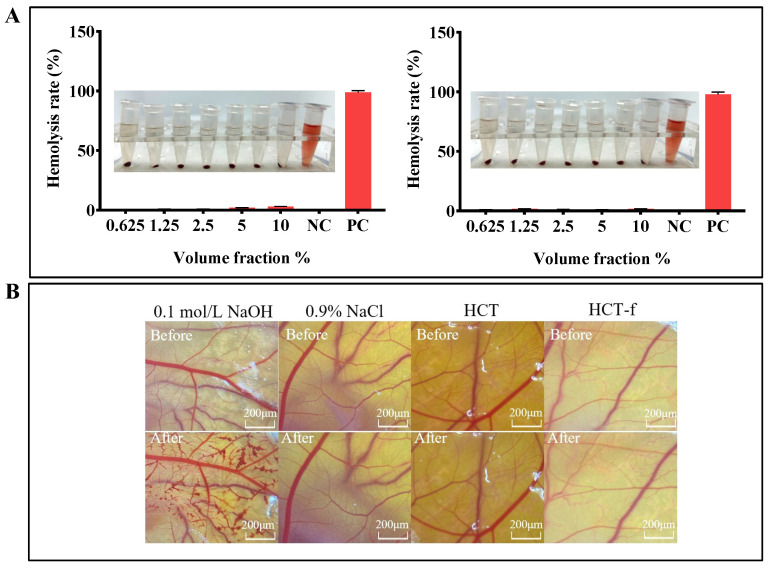
An exploration of the safety of HCT and HCT-f. (**A**) Effects of HCT and HCT-f on hemolysis of rabbit erythrocytes. (**B**) Effects of HCT and HCT-f on vascular damage in the chorioallantoic membrane of the chick embryo.

**Table 1 foods-13-01470-t001:** DPPH spiking sequence.

Reagent	Volume (mL)		
	A_1_	A_2_	A_3_
DPPH solution	1.0	1.0	0.0
Sample	1.0	0.0	1.0
Deionized water	0.0	1.0	1.0

**Table 2 foods-13-01470-t002:** Hydroxyl radical spiking sequence.

Reagent	Volume (mL)		
	A_0_	A_1_	A_2_
FeSO_4_ (8 mmol/L)	0.3	0.3	0.3
Salicylic acid (3 mmol/L)	1.0	1.0	0.0
Sample	0.0	1.0	1.0
Deionized water	1.45	0.45	1.45
H_2_O_2_ (0.02 mol/L)	0.25	0.25	0.25

**Table 3 foods-13-01470-t003:** Reverse transcription system.

Component Name	Volume (μL)
Anchored Oligo (dt) 18 Primer	1.0
2 × ES reaction Mix	10.0
Total RNA	2.0
EasyScript RT/RI Enzyme Mix	1.0
Gdna Remover	1.0
Rnase-free Water	5.0

**Table 4 foods-13-01470-t004:** Primer sequences.

Gene	Direction	Primer (5′-3′)
IL-6	F	TTCTCCACAAGCGCCTTC
	R	AGAGGTGAGTGGCTGTCTGT
IL-8	F	GGAGAAGTTTTTGAAGAGGGCTG
	R	ACAGACCCACACAATACATGAAG
TNF-α	F	TCTCCTTCCTGATCGTGGCA
	R	CAGCTTGAGGGTTTGCTACAAC
iNOS	F	CGTGGAGACGGGAAAGAAGT
	R	GACCCCAGGCAAGATTTGGA
AQP3	F	CTTCTTTGACCAGGACCGGC
	R	GGGCCAGCTTCACATTCTCT
FLG	F	TGAGGCATACCCAGAGGACT
	R	CTGTATCGCGGTGAGAGGAT
LOR	F	CTTCCTGGTGCTTTGGGCTC
	R	CTGGGGGATCTATTTGGACGG
Caspase-14	F	ATTCCACGGTAGAGGGATACA
	R	TCAGGGTTCGTTTTCCTTGCT
Caspase-3	F	ACTCCACAGCACCTGGTTAT
	R	TCTGTTGCCACCTTTCGGTT
Caspase-9	F	GGTGGACATTGGTTCTGGCAGAG
	R	ACGTTGTTGATGATGAGGCAGTGG
JNK	F	ATTGTTTGTGCTGCGTTCG
	R	CTTTTTGCGTGGGGTTGATT
ERK	F	CATCGCGACCTCAAACCTTC
	R	TCCGGATCTGCAACACGAG
NF-κB	F	CCAGGTCCTCCAGGCAATCCAAAAG
	R	AGTGCAGGGCTGTCAAACCATCGTAG
P38	F	ATGTTCCCTGTCCCTGAGC
	R	GCCCTGAGTGTAGCCTGTGT
Beta-actin	F	CTGAAGCCCCACTCAATCCA
	R	GCCAAGTCAAGACGGAGGAT

**Table 5 foods-13-01470-t005:** PCR reaction system.

Reagent	Volume (μL)
Template	1.5
Forward Primer (10 μM)	0.4
Reverse Primer (10 μM)	0.4
2 × TransStart^®^ Top Green qPCR SuperMix	10.0
Passive Reference Dye (50×)	0.4
Nuclease-free Water	7.3

**Table 6 foods-13-01470-t006:** Observation scoring record table.

Category	Score	Level	Performance
hemorrhage	0	None	-
	1	Mildly	Bleeding from small blood vessels and small amounts of hemorrhage only
	2	Moderate	Bleeding from small and large blood vessels with a noticeable amount of blood coming out
	3	Severe	Bleeding from almost all the blood vessels, a lot of blood coming out
coagulation	0	None	-
	1	Mildly	Mild intra/extravascular coagulation, mild clouding of CAM membranes
	2	Moderate	Moderate intra/extravascular coagulation with mild clouding of CAM membranes
	3	Severe	Heavy intra/extravascular coagulation with mild clouding of CAM membranes
angiolysis	0	None	-
	1	Mildly	Small vessel lysis only
	2	Moderate	Small and large vessel lysis
	3	Severe	Lysis of large vessels and the entire vascular tree

**Table 7 foods-13-01470-t007:** ES scoring standards.

Score Range	Irritation Classification
<6	Non-irritating
6 ≤ S ≤ 12	Mildly irritating
12 ≤ S ≤ 15	Moderately irritating
≥15	Severe irritation

**Table 8 foods-13-01470-t008:** Determination of OB and OBF molecular weights.

	HCT	HCT-f
Mn (Da)	848	1443
Mw (Da)	1.02758 × 10^5^	2.64265 × 10^5^
Mz (Da)	1.807968 × 10^6^	4.12595 × 10^6^
Mw/Mn	121.12684	183.09849
Mz/Mw	17.59448	15.61292

Mn: Number-average molecular weight; Mw: weight-average molecular weight; Mz: Z-average molecular weight.

**Table 9 foods-13-01470-t009:** ES scale for CAM bleeding after HCT and HCT-f treatment.

Sample	Bleeding Type	Rating/Point						Effective Score (ES)
		1	2	3	4	5	6	
0.9% NaCl solution	Hemorrhage	0	0	0	0	0	0	0
	Coagulation	0	0	0	0	0	0	
	Dissolution	0	0	0	0	0	0	
0.1 mol/L NaOH	Hemorrhage	2	2	2	2	2	2	12
	Coagulation	2	1	1	1	2	1	
	Dissolution	1	1	1	1	1	1	
HCT	Hemorrhage	0	0	0	0	0	0	0
	Coagulation	0	0	0	0	0	0	
	Dissolution	0	0	0	0	0	0	
HCT-f	Hemorrhage	0	0	0	0	0	0	0
	Coagulation	0	0	0	0	0	0	
	Dissolution	0	0	0	0	0	0	

## Data Availability

The original contributions presented in the study are included in the article, further inquiries can be directed to the corresponding author.

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
