# Peer review of "Inhibition of LPS-Induced Skin Inflammatory Response and Barrier Damage via MAPK/NF-κB Signaling Pathway by Houttuynia cordata Thunb Fermentation Broth"

_foods, 2024, doi:10.3390/foods13101470_

Round 1

Reviewer 1 Report

Comments and Suggestions for Authors

This article provides a comprehensive investigation into the effects of HCT (Houttuynia cordata Thunb) and its fermentation broth (HCT-f) on LPS-induced damage in HaCaT cells, focusing on oxidative stress, apoptosis, inflammatory factor release, skin barrier function, and the transcriptional activity of genes in the MAPK/NF-κB pathway.

While the study presents valuable insights into the potential benefits of HCT-f, there are some potential limitations in the article:

The article does not explicitly state the hypothesis or research questions guiding the study. Clear articulation of the research objectives at the beginning of the article would help readers understand the purpose of the study.

While the article mentions that statistical analysis was performed, it does not provide specific statistical tests used, degrees of freedom, p-values, or confidence intervals. Providing this information would enhance the credibility and replicability of the study.

The article could benefit from providing more background information on Houttuynia cordata Thunb, Clavispora lusitaniae, and their fermentation process. This would help readers understand the significance of the research within the broader scientific context.

Some data, particularly in the Results section, are presented in a tabular format without sufficient explanation or interpretation in the text. It would be more effective to integrate these findings into the narrative to enhance clarity and readability.

While some data are presented graphically, there could be much better graphs or figures to visually illustrate key findings, trends, or comparisons, which would aid in understanding complex results.

The article lacks sufficient detail in describing the experimental procedures, particularly in the Materials and Methods section. Providing more comprehensive information would ensure the reproducibility of the study by other researchers.

The article does not discuss any limitations or potential biases in the experimental design, methodology, or interpretation of results. Addressing these aspects could enhance the credibility and transparency of the study.

The article lacks contextualization of the findings within the broader scientific literature. Providing comparisons with existing studies or discussing how the results contribute to existing knowledge would strengthen the significance of the findings.

While the results suggest the potential benefits of HCT-f in mitigating LPS-induced damage, the study would benefit from additional validation through in vivo experiments or clinical trials to confirm the efficacy and safety of HCT-f in practical applications.

While the article explores the effects of HCT-f on various cellular processes, such as oxidative stress and apoptosis, the mechanistic insights remain somewhat unclear. Further elucidation of the underlying molecular mechanisms would enhance the understanding of how HCT-f exerts its effects.

The study does not address potential confounding factors or alternative explanations for the observed effects of HCT-f. Considering and controlling for confounders could strengthen the robustness of the findings.

The study primarily focuses on HaCaT cells and may not fully capture the complexity of in vivo physiological responses. Discussing the generalizability of the findings to other cell types or biological systems would provide a more comprehensive perspective.

Reproducibility is essential for scientific rigor. Providing detailed protocols and ensuring that the experiments can be replicated by other researchers would enhance the credibility and reliability of the findings.

The article lacks a detailed discussion section where the findings are interpreted in the context of existing literature and the broader implications of the results are discussed. A discussion section is crucial for providing insights, comparing findings with prior studies, and addressing limitations.

Addressing these points would contribute to the overall quality and reliability of the study, strengthening its contribution to the scientific understanding of HCT-f and its potential applications in mitigating skin inflammation and damage.

Comments on the Quality of English Language

Moderate editing of English language required

Author Response

Thank you very much for giving us this opportunity to revise our manuscript. Your comments and suggestions are of great help in improving the quality of our manuscripts. We have added the word file of the peer-to-peer responses as an attachment.

Reviewer 2 Report

Comments and Suggestions for Authors

The article entitled “Inhibition of LPS-induced skin inflammatory response and barrier damage via MAPK/NF-κB signaling pathway by Houttuynia cordata Thunb fermentation broth” deals with the fermentation broth of Houttuynia cordata Thunb and its reparative effect on inflammation in human immortalized keratino-cytes with lipopolysaccharide-induced injury.

The aims of the study are clearly expressed.

The experimental program is described in such manner that, with some amendments, it can be easily applied.

The obtained results are concise, presented and discussed.

Conclusions are drawn according to the obtained data.

The authors will find bellow some corrections and adjustments that should be addressed.

-          It is recommended to add full information about names, providers, countries for all the reagents and apparatus used in the experimental program. For many of the chemicals (such as Na2CO3 or gallic acid, DPPH, FeSO4 etc. for instance) these data are missing. It is not very clear how were some of the working conditions ensured (What apparatus was used for sterilization? How the stirring was insured? How the fermentation temperature was maintained? How were OD517, OD510 determined?)

-          Even though they are well-known, the abbreviation should be listed with their complete meaning at the first mention in the text (e.g. SEM does not have an explanation in the text, GPC-LS-IR is only partially explained)

-          The Latin names of the source used for the fermentation broth should be written with italic letters in all the manuscript.

-          Clavispora lusitaniae was used as inoculum for the fermentation step. Why was this yeast chosen? Is there other possibility of conducting the fermentation? With what results?

-          It is recommended to include in the „Results and Discussion” sections a comparison with fermentation broths obtained from other sources in order to highlight the importance of the chosen Houttuynia cordata Thunb.

Comments on the Quality of English Language

Minor editing of English language required

Author Response

(The authors gave the same response as above.)

Reviewer 3 Report

Comments and Suggestions for Authors

1-The abstract must be rewritten, the methods not clear, and the numerical results must  be added

2-line 106 the word (dissolved) do not appropriate

3-Line 107. Where was Clavispora lusitaniae, was obtained?, what was the amount added to the media, what the conditions, is there any nutrients were added to enhance fermentations

4-Please added references to the part of the determination of Phenol

5-part cell culture and viability, please accurately define each group 

6- you wrote different concentrations in the result only one concentration and added the concentrations used with the cell line

7-please do not start any sentence with a number

8-Table 8 please add the abbreviation

9- Figure 4a,b the variability of control 100% did not add HCT or HCT.F, please add one column, and how the viability % exceeds 100% how to calculate

10. the images in figures 5,6 not clear please add clear photo

Author Response

(The authors gave the same response as above.)

Round 2

Reviewer 3 Report

Comments and Suggestions for Authors

Accept in present form